# GPT AS VISUAL EXPLAINER

## ABSTRACT

In this paper, we present Language Model as Visual Explainer (`LVX`), a systematic approach for interpreting the internal workings of vision models using a tree-structured linguistic explanation, without the need for model training. Central to our strategy is the collaboration between vision models and LLM to craft explanation. On one hand, the LLM is harnessed to delineate hierarchical visual attributes, while concurrently, a text-to-image API retrieves images that are most align with these textual concepts. By mapping the collected text and image to the vision model's embedding space, we construct a hierarchy-structured visual embedding tree. This tree is dynamically pruned and grown by querying the LLM using language templates, tailoring the explanation to the model. Such a scheme allows us to seamlessly incorporate new attributes while eliminating undesired concepts based on the model's representations. When applied to testing samples, our method provides human-understandable explanations in the form of attribute-laden trees. Beyond explanation, we retrained the vision model by calibrating the model on the generated concept hierarchy, allowing the model to incorporate the refined knowledge of visual attributes. To access the effectiveness of our approach, we introduce new benchmarks and conduct rigorous evaluations. The results unequivocally demonstrate the plausibility, faithfulness, and stability of our approach compared to existing interpretability techniques.

## 1 INTRODUCTION

Unlocking the secrets of deep neural networks is akin to navigating through an intricate, ever-shifting maze, as the intricate decision flow within the networks is, in many cases, extremely difficult for humans to fully interpret. As we delve deeper into safety-critical domains like medical applications and autonomous driving, the lack of interpretability and the presence of uncertainty pose significant obstacles to users who need to trust the decisions made by these systems. In this quest, extracting clear, understandable explanations from these perplexing mazes has become an imperative task.

While efforts have been developed to address the lack of explainability in the domain of computer vision, these approaches often fall short of providing direct and human-understandable explanations. Standard techniques, such as attribution methods [Lundberg & Lee, 2017; Ribeiro et al., 2016; Zeiler & Fergus, 2014; Smilkov et al., 2017], feature importance [Selvaraju et al., 2017; Simonyan et al., 2013; Shrikumar et al., 2017] and prototype analysis [Chen et al., 2019; Nauta et al., 2021], only highlight certain pixels or features that are deemed important by the model. As such, these methods often require the involvement of experts to verify or interpret the outputs for non-technical users. Natural language explanations [Hendricks et al., 2016; Camburu et al., 2018; Li et al., 2018; Kim et al., 2018], on the other hand, present an attractive alternative, since the produced texts are better aligned with human cognition. Nevertheless, these approaches typically rely on labor-intensive and biased manual annotation of textual rationales for model training.

In this study, we take a bold step toward bridging the gap between human comprehension and AI decision. We present a systematic approach, Language Model as Visual Explainer (`LVX`), for interpreting vision models using tree-structured language explanations, without model training. The primary challenge we face is that models trained solely on pixel data lack an understanding of the visual concepts present in an image. For example, if a model predicts one image as a "dog", it is unclear whether it truly recognizes the features like the *wet nose* or *floppy ear*, or if it is merely making irrational guesses. To address this challenge, we propose linking the visual model with a robust, external knowledge provider to establish connections between visual attributes and image

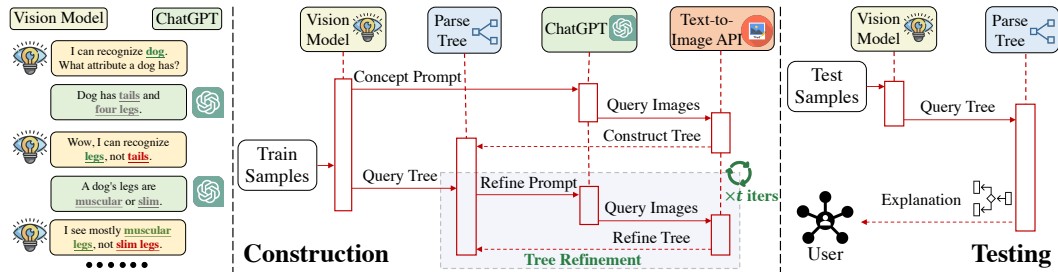

Figure 1: General workflow of LVX. (Left) A toy example that LLM interacts with vision model to examine its capability. (Mid) It combines vision, language, and visual-language APIs to create a parse tree for each visual model. (Right) In testing, embeddings navigate this tree, and the traversed path provides a personalized explanation for the model's prediction.

patterns. To this end, we leverage large language models (LLM) such as ChatGPT and GPT4 as our knowledge providers, combining them with the visual recognition system. Figure 1 (Left) describes a toy case, where the LLM is interacts with the vision model to explore its capability boundaries. By doing so, we gain insights into the what visual attributes can be recognized by the model.

The pipeline of our approach is illustrated in Figure 1, which comprises two main stages, the *construction phase* and the *test phase*.

In the *construction phase*, our goal is to create an attribute parse tree for each category, partitioning the feature space of a visual model via LLM-defined attribute hierarchy. We begin by extracting commonsense knowledge about each category and its visual attributes from LLMs using in-context prompting [Liu et al., 2021] and organize this knowledge into a tree. Utilizing a text-to-image API, we gather corresponding images. These are subsequently inputted into the vision model to extract prototype embeddings, which are then mapped to the tree nodes. The parse tree is then refined based on the properties of the training set. Each training sample is processed by the vision model, with its features navigating the parse tree based on their proximity to prototype embeddings. Infrequently visited nodes, representing attributes less recognizable by the model, are pruned. Conversely, nodes that are visited often, signifying that the model can efficiently recognize the associated concepts, induce the tree's growth as the LLM introduces refined concepts. Consequently, LVX yields human-understandable attribute trees that mirror the model's understanding of each concept.

In the *test phase*, we input a test sample into the model to extract its feature. The feature is then routed in the parse tree by finding the nearest neighbor at each node. The path from the root to the leaf node serves as a sample-specific rationale for the model's prediction, offering an explanation of how the model arrived at its decision.

To validate our approach's efficacy, we collect new hierarchical annotations and design new metrics to evaluate the performance of LVX on various real-world datasets. Notably, the dataset is solely used for *evaluation purposes*. Beyond interpretation, our study proposes to calibrate the vision model by utilizing the generated explanation results. By leveraging insights obtained from the tree-structured explanations, we can improve the model's performance, leading to reliable decision-making processes. Experimental results demonstrate the effectiveness of our method compared to existing interpretability techniques, highlighting its potential for advancing explainable AI.

To summarize, Our main contributions are:

- The paper introduces a novel task, visual explanatory tree parsing, that interprets vision models using tree-structured language explanations.

- We introduce the Language Model as Visual Explainer (LVX) to carry out the visual explanatory tree parsing task without model training. The proposed LVX is the first dedicated approach to leverage language models to explain the visual recognition system.

- Our study proposes leveraging the generated explanation results to calibrate the vision model, leading to enhanced performance and improved reliability in decision-making processes.

- To validate the credibility of our tree-structured explanations, we present several new benchmarks and design new metrics, facilitating rigorous evaluation of the LVX method's plausibility, faithfulness, and stability on real-world datasets.

## 2    PROBLEM DEFINITION

We first define our specialized task, called **visual explanatory tree parsing**, which seeks to unravel the decision-making process of a vision model through a tree. Let us consider the trained vision model $f$, defined as a function $f\colon \mathcal{X} \to \mathcal{Y}$, where $\mathcal{X}$ represents the input image space and $\mathcal{Y}$ denotes the output label space. In this study, our focus lies on the classification task, where $f = g \circ h$ is decomposed into a feature extractor $g$

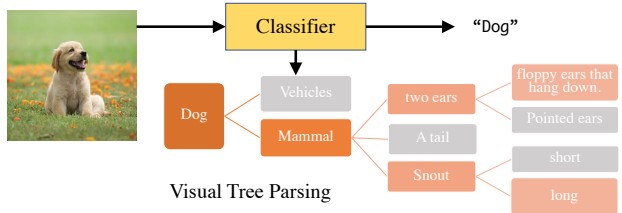

Figure 2: The illustration of visual explanatory tree parsing task, where each input sample is interpreted as a parse tree to represent the model's logical process.

and a linear classification head $h$. The output space $\mathcal{Y} \in \mathbb{R}^n$, where $n$ signifies the number of classes. The model is trained on a labeled training set $D_{tr} = \{\mathbf{x}_j, y_j\}_{j=1}^M$ would be evaluated a test set $D_{ts} = \{\mathbf{x}_j\}_{j=1}^L$.

The ultimate objective of our problem is to generate an explanation $T$ for each model-input pair $(f, \mathbf{x})$ on the test set, illuminating the reasoning behind the model's prediction $\hat{y} = f(\mathbf{x})$. This unique explanation manifests as a tree of attributes, denoted as $T = (V, E)$, comprising a set of $L$ nodes $V = \{v_i\}_{i=1}^{N_v}$ and $M$ edges $E = \{e_i\}_{i=1}^{N_e}$. The root of the tree is the predicted category, $\hat{y}$, while each node $v_i$ encapsulates a specific attribute description of the object. These attributes are meticulously organized, progressing from the holistic to the granular, and from the general to the specific.

Figure 2 provides a visual illustration of our proposed task. Unlike existing approaches [Radford et al., 2021; Alayrac et al., 2022] that rely on visual-language paired training [Menon & Vondrick, 2022; Mao et al., 2022; Pellegrini et al., 2023; Yang et al., 2023; Zhang et al., 2023], we address the more challenging scenario, on explaining vision models trained solely on pixel data. While some models can dissect and explain hierarchical clustering of feature embeddings [Singh et al., 2019; Wan et al., 2020], they lack the ability to associate each node with a textual attribute. It is important to note that our explanations primarily focus on examining the properties of the established network, going beyond training model for reasoning hierarchy [Feinerer & Hornik, 2023] and attributes [Isola et al., 2015] from the image. Notably, our approach achieves this objective *without supervision*, eliminating the need for predefined hierarchical ground truth explanations for model training.

## 3    LANGUAGE MODEL AS VISUAL EXPLAINER

This section dives deep into the specifics of LVX. At the heart of our approach is the interaction between the LLM and the vision model to construct the parsing tree. Subsequently, we establish a rule to route through these tree, enabling the creation of coherent text explanations.

### 3.1    TREE CONSTRUCTION VIA LLM

Before constructing our trees, let's take a moment to examine how humans accomplish this task. Typically, we already hold a hierarchy of concepts in our minds. When presented with visual stimuli, we instinctively compare the data to our existing knowledge tree, confirming the presence of distinct traits. We effortlessly recognize familiar traits and, for unfamiliar ones, we expand our mental framework. For example, when we think of a dog, we typically know that it has a *furry tail*. Upon observing a dog, we naturally check for the visibility of its tail. If we encounter a *hairless tail*, previously unknown to us, we incorporate it into our knowledge base, ready to apply it to other dogs.

Our LVX mirrors this methodology. We employ LLM as a "knowledge provider" to construct the initial conceptual tree. Subsequently, we navigate through the visual model's feature space to assess the prevalence of each conceptual node. If a specific attribute is rarely observed, we remove the corresponding nodes from the tree. Conversely, if the model consistently recognizes an attribute, we enrich the tree by integrating more nuanced, next-level concepts. This iterative process ensures the refinement and adaptation of the conceptual tree within our pipeline, which gives rise to our LVX.

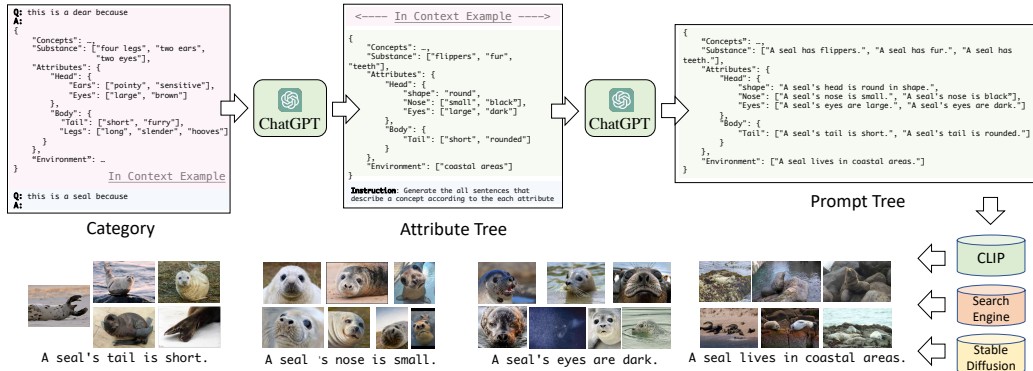

Figure 3: Crafting text-image pairs for visual concepts. Through in-context prompting, we extract knowledge from the LLM, yielding visual attributes for each category. These attributes guide the collection of text-image pairs that encapsulate the essence of each visual concept.

**Generating Textual Descriptions for Visual Concepts.** We leverage a large language model (LLM) as our "commonsense knowledge provider" [Li et al., 2022; Zhou et al., 2020] to generate textual descriptions of visual attributes corresponding to each category. The LLM acts as an external database, providing a rich source of diverse visual concept descriptions. The process is illustrated in Figure 3.

Formally, assume we have a set of category names, denoted as $C = \{c_i\}_{i=1}^n$, where $i$ represents the class index. For each of these classes, we prompt an LLM $L$ to produce visual attribute tree. We represent these attributes as $d_i = L(c_i, \mathcal{P})$, where $d_i$ is a nested JSON text containing textual descriptions associated with class $c_i$. To help generate $d_i$, we use example input-output pairs, $\mathcal{P}$, as in-context prompts. The process unfolds in two stages:

- **Initial Attribute Generation**: We initially generate keywords that embody the attributes of each class. The contextual prompt follows a predefined template that instructs the LLM to elaborate on the attributes of a visual object. The template is phrased as "This is a <CLSNAME> because". The output JSON contains four primary nodes: Concepts, Substances, Attributes, and Environments. As such, the LLM is prompted to return the structured key attributes of a visual object. Note that the initial attributes tree may not accurately represent the model; refinements will be made in the refinement stage.

- **Description Composition**: Next, we guide the LLM to create descriptions based on these attributes. Again we showcase an in-context example and instruct the model to output "Generate sentences that describe a concept according to each attribute."

Once the LLM generates the structured attributes $d_i$, we parse them into an initial tree, represented as $T_i^{(0)} = (V_i^{(0)}, E_i^{(0)})$, using the key-value pairs of the JSON text. Those generated JSON tree is then utilized to query images corresponding to each factor.

**Visual Embeddings Tree from Retrieved Images.** In order to enable the vision model to understand attributes generated by the LLM, we employ a two-step approach. The primary step involves the conversion of textual descriptions, outputted by the LLM, into images. Then, these images are deployed to investigate the feature region that symbolizes specific attributes within the model.

The transition from linguistic elements to images is facilitated by the use of arbitrary text-to-image API. This instrumental API enables the generation of novel images or retrieval of existing images that bear strong relevance to the corresponding textual descriptions. An initial parse tree node, denoted by $v$, containing a textual attribute, is inputted into the API to yield a corresponding set of $K$ support images, represented as $\{\widetilde{\mathbf{x}}_i\}_{i=1}^K = \text{T2I}(v)$. The value of $K$ is confined to a moderately small range, typically between 5 to 30. The full information of the collected dataset will be introduced in Section 4.

Our research incorporates the use of search engines such as Bing, or text-to-image diffusion models like Stable-Diffusion [Rombach et al., 2021], to derive images that correspond accurately to the provided attributes.

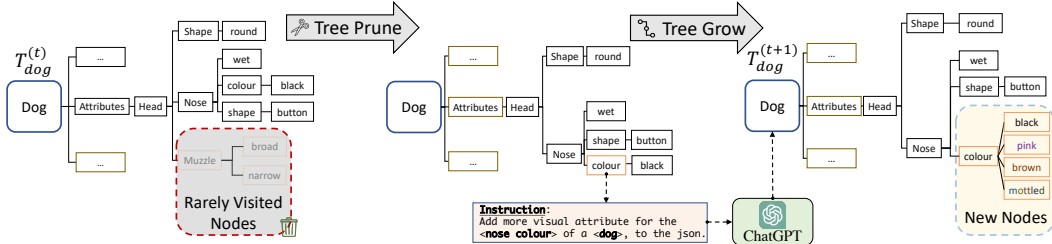

Figure 4: Tree refinement by traversing the embedding tree and querying the LLM model.

Following this, the images are presented to the visual model to extract their respective embeddings, represented as $\mathbf{p}_i = g(\widetilde{\mathbf{x}}_i)$. As such, each tree node contains a set of support visual features $P = \{\mathbf{p}_k\}_{k=1}^{K}$. This procedure allows for the construction of an embedding tree, consisting of paired text and visual features. These pairs are arranged in a concept tree structure as prescribed by the LLM. It is important to note that the collected images are not employed in training the model. Instead, they serve as a support set to assist the model in understanding and representing the disentangled attributes effectively. As such, the visual model uses these embeddings as a map to navigate through the vast feature space, carving out territories of attributes, and laying down the groundwork for further exploration and explanation of a particular input.

**Tree Refinement Via Refine Prompt.** Upon construction, the parse tree structure is refined to better align with the model's feature spaces. This stage, termed *Tree Refinement*, is achieved through passing training data as a query to traverse the tree. Nodes that are seldom visited indicate that the model infrequently recognizes their associated attributes. Therefore, we propose a pruning mechanism that selectively eliminates these attributes, streamlining the tree structure. For nodes that frequently appear during the traversal, we further grow the tree by introducing additional or more detailed attributes, enriching the overall context and depth of the tree. The procedure is demonstrated in Figure 4.

Initially, we treat the original training samples, denoted as $(\mathbf{x}_j, y_j) \in D_{tr}$, as our query set. Each sample is passed to the visual model to extract a feature, represented as $\mathbf{q}_j = g(\mathbf{x}_j)$.

Next, the extracted feature traverses the tree corresponding to $y_j$ from the root. Its aim is to locate the closest semantic neighbor among the tree nodes. We define a distance metric between $\mathbf{q}_j$ to support set $P$ as the point-to-set distance $D(\mathbf{q}_j, P)$. This metric represents the greatest lower bound of the set of distances from $\mathbf{q}_j$ to prototypes in $P$. Therefore, the distance metric is given by:

$$D(\mathbf{q}_j, P) = \inf\{d(\mathbf{q}_j, \mathbf{p})|\mathbf{p} \in P\} \qquad (1)$$

where $d(\cdot, \cdot)$ denotes the Euclidean distance. Following this, we employ a Depth-First Search (DFS) algorithm to locate the tree node closest to the query point $\mathbf{q}_j$. After finding this node, each training point $(\mathbf{x}_j, y_j)$ is assigned to a specific node of the tree. Subsequently, we count the number of samples assigned to a particular node $v^*$, using the following formula:

$$C_{v^*} = \sum_{j=1}^{M} \mathbb{1}\{v^* = \underset{v \in V_{y_j}^{(0)}}{\arg\min} D(\mathbf{q}_j, P_v)\} \qquad (2)$$

In this formula, $\mathbb{1}$ is the indicator function and $P_v$ denotes the support feature for node $v$. Following this, we rank each node based on the sample counter, which results in two operations to update the tree architecture $T_i^{(t+1)} = \texttt{Grow}(\texttt{Prune}(T_i^{(t)}))$, where $t$ stands as the iteration number

- **True Pruning**. Nodes with the least visits are pruned from the tree, along with their child nodes.

- **True Growing**. For the top-ranked node, we construct a new inquiry to prompt the LLM to generate attributes with finer granularity. The inquiry is constructed with an instruction template "Add visual attributes for the <NODENAME> of a <CLASSNAME>, to the json".

The revised concept tree generated by the LLM provides a comprehensive and detailed representation of the visual attribute. To refine the attribute further, we employ an iterative procedure that involves image retrieval and the extraction of visual embeddings, as illustrated in Figure 1. This iterative process enhances the parse tree by incorporating new elements. As each new element is introduced, the attribute areas within the feature space become increasingly refined, leading to improved interpretability of the model. In our experiment, we performed five rounds of tree refinement.

## 3.2 ROUTING IN THE TREE

Once the tree is established, the model predicts the class of a new test sample $\mathbf{x}'$ and provides an explanation for this decision by finding the top-k nearest neighbor nodes.

Specifically, the model predicts the category $\hat{y}$ for the object in the test sample $\mathbf{x}'$ as $\hat{y} = f(\mathbf{x}')$. The extracted image feature $\mathbf{q}'$ corresponding to $\mathbf{x}'$ is routed through the tree. Starting from the root, the tree is traversed to select the top-k nearest neighbor nodes $\{v_i\}_{i=1}^{k}$ based on the smallest $D(\mathbf{q}', P_{v_i})$ values, representing the highest semantic similarity between $\mathbf{q}'$ and the visual features in the tree's nodes. The paths from the root to the selected nodes are merged to construct the explanatory tree $T$ for the model's prediction.

This parse tree structure reveals the sequence of visual attributes that influenced the model's classification of $\mathbf{x}'$ as $\hat{y}$. It facilitates the creation of precise, tree-structured justifications for these predictions. Importantly, the routing process involves only a few feature similarity computations per node and does not require queries to the large language model,resulting in exceptionally fast processing.

## 3.3 CALIBRATING THROUGH EXPLAINING

The created parse tree offers a two-fold advantage. Not only does it illustrate the logic of a specific prediction, but it also serves as a by-product to refine the model's predictions by introducing hierarchical regularization for learned representation. Our goal is to use the parse tree's explanations as pseudo-labels, embedding this hierarchical knowledge into the model.

To operationalize this, we employ a hierarchical multi-label contrastive loss (HiMulCon) Zhang et al. [2022], denoted as $\mathcal{L}_{HMC}$, to fine-tune the pre-trained neural network. This approach enhances the model by infusing structured explanations into the learning process, thus enriching the representation.

Specifically, we apply the LVX on all training samples. The explanatory path $\hat{T}_j$ provides a hierarchical annotation for each training sample $\mathbf{x}_j$. The model is trained with both the cross-entropy loss $\mathcal{L}_{CE}$ and $\mathcal{L}_{HMC}$ as follows:

$$\min \sum_{j=1}^{M} \mathcal{L}_{CE}\Big(f(\mathbf{x}_j), y_j\Big) + \lambda \mathcal{L}_{HMC}\Big(g(\mathbf{x}_j), \hat{T}_j\Big) \tag{3}$$

Here, $\lambda$ is a weighting coefficient. The explanation $\hat{T}_j$ is updated every 10 training epochs to ensure alignment with the network's evolving parameters and learning progress. Notably, the support set isn't used in model training, maintaining a fair comparison with the baselines.

## 4 EXPERIMENT

This section offers an in-depth exploration of our evaluation process for the proposed LVX framework and explains how it can be utilized to gain insights into the behavior of a trained visual recognition model, potentially leading to performance and transparency improvements.

## 4.1 EXPERIMENTAL SETUP

**Data Annotation and Collection.** To assess explanation plausibility, data must include human annotations. Currently, no large-scale vision dataset with hierarchical annotations is available to facilitate reasoning for visual predictions. To address this, we developed annotations for three recognized benchmarks: CIFAR10, CIFAR100 [Krizhevsky, 2009], and ImageNet Russakovsky et al. [2015], termed as H-CIFAR10, H-CIFAR100, and H-ImageNet. These annotations, detailed in Table 1, serve as ground truth for model evaluation, highlighting our dataset's unique support for hierarchical attributes and diverse visual concepts. Note that, we evaluate on hierarchical datasets only, as our method is specifically designed for structured explanations.

As an additional outcome of our framework, we have gathered three support sets to facilitate model explanation. In these datasets, each attribute generated by the LLM corresponds to a collection of images that showcase the specified visual concepts. These images are either retrieved from Bing search engine [1] using attributes as queries or are generated using Stable-diffusion. We subsequently

---
[1] https://www.bing.com/images/

Table 1: Data annotation statistics. The $*$ indicates the number of video frames. We compare the statistics of category, attributes, image and tree depth across different explanatory datasets. Our dataset stands out as the first hierarchical dataset, offering a wide range of attributes.

| Dataset Name | No. Categories | No. Attributes | No. Images | Avg. Tree Depth | Rationales | Hierarchy | Validation Only |
|---|---|---|---|---|---|---|---|
| AWA2 Xian et al. [2018] | 50 | 85 | 37,322 | N/A | ✓ | ✗ | ✗ |
| CUB Wah et al. [2011] | 200 | N/A | 11,788 | N/A | ✓ | ✗ | ✗ |
| BDD-X Kim et al. [2018] | 906 | 1,668 | 26,000* | N/A | ✓ | ✗ | ✗ |
| VAW Pham et al. [2021] | N/A | 650 | 72,274 | N/A | ✗ | ✗ | ✗ |
| COCO Attr Patterson & Hays [2016] | 29 | 196 | 180,000 | N/A | ✗ | ✗ | ✗ |
| DR-CIFAR-10 Mao et al. [2022] | 10 | 63 | 2,201 | N/A | ✓ | ✗ | ✗ |
| DR-CIFAR-100 Mao et al. [2022] | 100 | 540 | 18,318 | N/A | ✓ | ✗ | ✗ |
| DR-ImageNet Mao et al. [2022] | 1,000 | 5,810 | 271,016 | N/A | ✓ | ✗ | ✗ |
| H-CIFAR-10 | 10 | 289 | 10,000 | 4.3 | ✓ | ✓ | ✓ |
| H-CIFAR-100 | 100 | 2,359 | 10,000 | 4.5 | ✓ | ✓ | ✓ |
| H-ImageNet | 1,000 | 26,928 | 50,000 | 4.8 | ✓ | ✓ | ✓ |

filter the mismatched pairs with the CLIP model, with the threshold of 0.5. Due to the page limit, extensive details on data collection, false positive removal, limitations, and additional evaluation on medical data, such as X-ray diagnoses, are available in the supplementary material.

**Evaluation Metrics.** In this paper, we evaluate the quality of our explanation from three perspectives: *Plausibility*, *Faithfulness* and *Stability*.

- **Plausibility** measures how reasonable the machine explanation is compared to the human explanation. For plausibility assessment, we leverage two conventional metrics for analyzing tree similarity: Maximum Common Subgraph (MCS) Raymond & Willett [2002]; Kann [1992], and Tree Kernels (TK) Sun et al. [2011]. We calculate their normalized scores respectively. Specifically, given a predicted tree $T_{pred}$ and the ground-truth $T_{gt}$, the MCS score is computed as $\frac{|MCS| \times 100}{\sqrt{|T_{pred}||T_{gt}|}}$, and the TK score is computed as $\frac{TK(T_{pred}, T_{gt}) \times 100}{\sqrt{TK(T_{pred}, T_{pred})TK(T_{gt}, T_{gt})}}$. Here, $|\cdot|$ represents the number of nodes in a tree, and $TK(\cdot, \cdot)$ denotes the unnormalized TK score. We report the average score across all validation samples.

- **Faithfulness** states that the explanations should reflect the inner working of the model. We introduce Model-induced Sample-Concept Distance (MSCD) to evaluate this, calculated as the average of point-to-set distances $\frac{1}{N_v} \sum_{v \in V} D(\mathbf{q}_j, P_v)$ between all test samples and tree nodes, reflecting the alignment between generated explanation and model's internal logic. The concept is simple: if the explanation tree aligns with the model's internal representation, the MSCD is minimized, indicating high faithfulness.

- **Stability** evaluates the resilience of the explanation graph to minor input variation, expecting minimal variations in explanations. The MCS/TK metrics are used to assess stability by comparing explanations derived from clean and slightly modified inputs. We include 3 perturbations, including Gaussian additive noise with $\sigma \in \{0.05, 0.1\}$ and Cutout [DeVries & Taylor, 2017] augmentation.

**Baselines.** Given the absence of pre-existing models capable of hierarchical explanations without supervision, we introduce baseline models: `Constant`, using the full category template tree; `Random`, which selects a subtree randomly from the template; and `Subtree`, choosing the most common subtree in the test set for explanations. Additionally, we consider `TrDec` Baseline [Wang et al., 2018], a strategy utilizing a tree-topology RNN decoder on top of image encoder. Given the absence of hierarchical annotations, the CLIP model verifies nodes in the template trees, serving as pseudo-labels for training. We only update the decoder parameters for interpretation purposes. These models provide a basic comparison for the performance of `LVX`. More details are in the appendix.

For classification performance, we compare `LVX`-calibrated model with to neural-tree based solutions, including a Decision Tree (DT) trained on the neural network's final layer, DNDF [Kontschieder et al., 2015], and NBDT [Wan et al., 2020].

**Models to be Explained.** Our experiments cover a wide range of supervised trained neural networks, including various convolutional neural networks (CNN) and transformer architectures. These models consist of VGG [Simonyan & Zisserman, 2014], ResNet [He et al., 2016], DenseNet [Huang et al., 2017], GoogLeNet [Szegedy et al., 2015], Inceptionv3 [Szegedy et al., 2016], MobileNet-v2 [Sandler et al., 2018], and Vision Transformer (ViT) [Dosovitskiy et al., 2020]. In total, we utilize 12 networks for CIFAR-10, 11 networks for CIFAR-100, and 8 networks for ImageNet. For each model, we perform the tree refinement for 5 iterations.

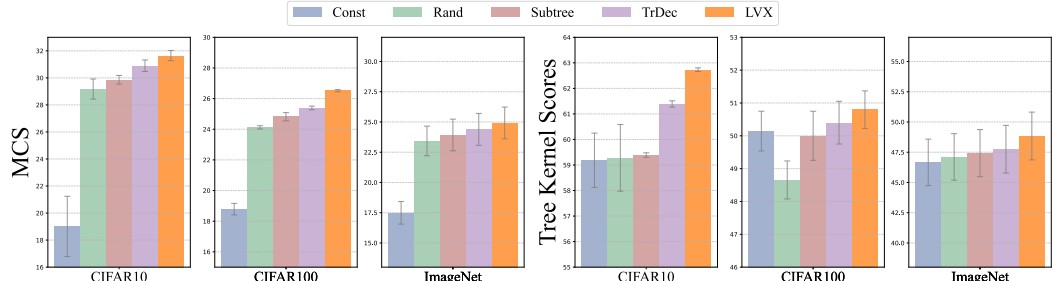

Figure 5: *Plausibility* comparison on three visual tree parsing benchmarks. We plot the mean±std across all networks architectures. For both scores, higher values indicate better performance.

| Method | Network | Clean | | Gaussian ($\sigma = 0.05$) | | Gaussian ($\sigma = 0.1$) | | Cutout ($n_{holes} = 1$) | |
|--------|---------|-----|-----|-----|-----|-----|-----|-----|-----|
| | | MCS | TK | MCS | TK | MCS | TK | MCS | TK |
| TrDec | RN-18 | 100 | 100 | 65.3 | 86.4 | 56.2 | 82.5 | 65.4 | 86.0 |
| LVX | RN-18 | 100 | 100 | **69.7** | **90.8** | **62.1** | **86.5** | **68.1** | **88.3** |
| TrDec | RN-50 | 100 | 100 | 68.3 | 88.5 | 59.3 | 84.2 | 66.2 | 86.9 |
| LVX | RN-50 | 100 | 100 | **71.9** | **92.1** | **65.6** | **88.3** | **69.3** | **90.1** |

| Network | CIFAR-10 | | | CIFAR-100 | | | ImageNet | | |
|---------|------|---------|-----|------|---------|-----|------|---------|-----|
| | TrDec | SubTree | LVX | TrDec | SubTree | LVX | TrDec | SubTree | LVX |
| RN-18 | 3.98 | 2.08 | **0.61** | 3.58 | 1.78 | **1.29** | 2.88 | 1.37 | **0.93** |
| RN-50 | 3.75 | 1.86 | **0.36** | 3.42 | 1.54 | **0.45** | 2.68 | 1.25 | **0.44** |
| ViT-S 16 | 3.62 | 1.68 | **0.23** | 3.28 | 1.44 | **0.52** | 2.56 | 1.15 | **0.20** |

Table 2: *Stability* comparison in CIFAR10 under input perturbations.

Table 3: *Faithfulness* comparison by computing the MSCD score. Smaller the better.

**Calibration Model Training.** As described in Section 3.3, we finetune the pre-trained neural networks with the hierarchical contrastive loss based on the explanatory results. The model is optimized with SGD for 50 epochs on the training sample, with an initial learning rate in $\{0.001, 0.01, 0.03\}$ and a momentum term of 0.9. The weighting factor is set to 0.1. We compare the calibrated model with the original model in terms of accuracy as well as the explanation performance.

## 4.2 LLM HELPS VISUAL INTERPREBILITY

**Plausibility Results.** We evaluated LVX against human annotations across three datasets, using different architectures, and calculating MCS and TK scores. The results, shown in Figure 5, reveal LVX outperforms baselines, providing superior explanations. Notably, TrDec, even when trained on CLIP induced labels, fails to generate valid attributes in deeper tree layers—a prevalent issue in long sequence and structure generation tasks. Meanwhile, SubTree lacks adaptability in its explanations, leading to lower scores. The insights each individual network are mentioned in the appendix.

**Faithfulness Results.** We present the MSCD scores for ResNet-18(RN-18), ResNet-50(RN-50), and ViT-S, contrasting them with SubTree and TrDec in Table 3. Thanks to the incorporation of tree refinement that explicitly minimizes MSCD, our LVX method consistently surpasses benchmarks, demonstrating lowest MSCD values, indicating its enhanced alignment with model reasoning.

**Stability Results.** The stability of our model against minor input perturbations on the CIFAR-10 dataset is showcased in Table 2, where MCS/TK are computed. The "Clean" serves as the oracle baseline. Our method, demonstrating robustness to input variations, retains consistent explanation results (MCS>60, TK>85). In contrast, TrDec, dependent on an RNN-parameterized decoder, exhibits higher sensitivity to feature variations.

**Model and Data Diagnosis with Explanation.** We visualize the sample explanatory parse tree on ImageNet validation set induced by ViT-B in Figure 6. The explanations fall into three categories: (1) correct predictions with explanations, (2) incorrect predictions with explanations, and (3) noisy label predictions with explanations. We've also displayed the 5 nearest neighbor node for each case.

What's remarkable about LVX is that, even when the model's prediction is wrong, it can identify correct attributes. For instance, in a case where a "white shark" was misidentified as a "killer whale" (b-Row 2), LVX correctly identified "fins", a shared attribute of both species. Moreover, the misrecognition of the attribute "wide tail flukes" indicates a potential error in the model, that could be later addressed to enhance its performance.

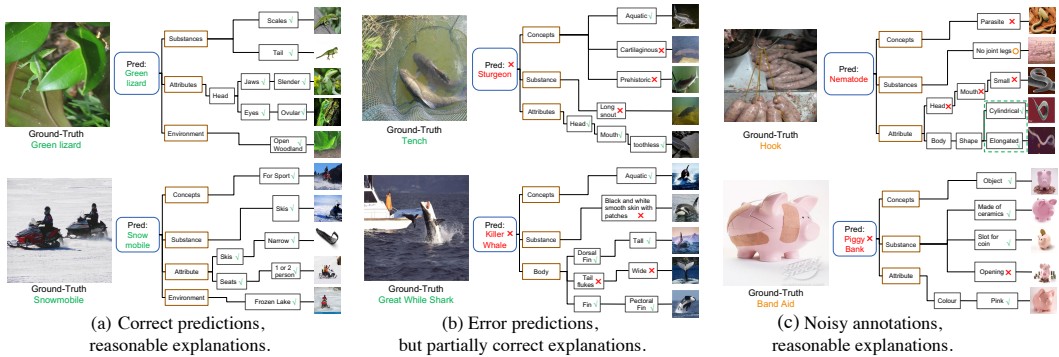

(a) Correct predictions, reasonable explanations.

(b) Error predictions, but partially correct explanations.

(c) Noisy annotations, reasonable explanations.

Figure 6: Explanation visualization for ViT-B on ImageNet-1K. ✓ and × means that the node is aligned or misaligned with the image. Zoom in for better view.

Surprisingly, `LVX` is able to identify certain noisy labels in the data, as shown in c-Row 2. In such cases, even experienced human observers might struggle to decide whether a "`pig bank with band`" should be classified "`piggy bank`" or "`band aid`". It again underscores the superior capabilities of our `LVX` system in diagnosing the errors beyond model, but also within the data itself.

**Calibration Enhances Interpretability and Performance.** Our approach involves fine-tuning a pre-trained model with the loss function outlined in Section 3.3, using parsed explanatory trees to improve model performance. Table 4 compares the classification performance of our model with that of other neural tree methods. Our model clearly outperforms the rest.

While neural tree models often struggle to balance between interpretability and performance, our `LVX` model circumvents this problem. `LVX` differs from traditional approaches in that, it doesn't necessitate an exact decision tree rule. Instead, the decision is made by the neural network, with knowledge from the LLM incorporated into the model via Equation.3. This promotes the model's capacity to disentangle visual concepts, yielding explainability and awareness of visual attributes, an edge that other models do not possess.

In addition, we compared the quality of the generated parsed tree with or without calibration, in Figure 7. The calibration process not only improved model performance, but also led to more precise tree predictions, indicating enhanced interpretability. We also test the calibrated model on OOD evaluations in Appendix, where we observe notable improvements.

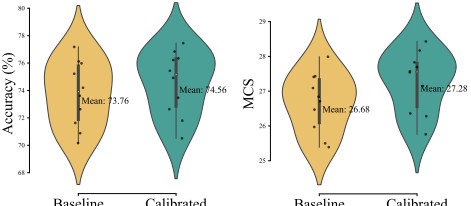

Figure 7: Performance and interpretability comparison with/without model calibration on CIFAR-100. Higher MCS means better.

| Method | Network | Expl. | CIFAR10 | CIFAR100 | ImageNet |
|---|---|---|---|---|---|
| NN | ResNet18 | × | 94.97% | 75.92% | 69.76% |
| DT | ResNet18 | ✓ | 93.97% | 64.45% | 63.45% |
| DNDF | ResNet18 | ✓ | 94.32% | 67.18% | N/A |
| NBDT | ResNet18 | ✓ | 94.82% | 77.09% | 65.27% |
| LVX (Ours) | ResNet18 | ✓ | **95.14%** | **77.33%** | **70.28%** |

Table 4: Performance comparison of neural decision tree-based methods. *Expl.* stands for whether the prediction is explainable.

## 5 CONCLUSION

In this study, we introduced `LVX`, an approach for interpreting vision models using tree-structured language explanations without hierarchical annotations. `LVX` leverages large language models to connect visual attributes with image features, generating comprehensive explanations. We refined attribute parse trees based on the model's recognition capabilities, creating human-understandable descriptions. Test samples were routed through the parse tree to generate sample-specific rationales. `LVX` demonstrated effectiveness in interpreting vision models, offering potential for model calibration. Our contributions include proposing `LVX` as the first approach to leverage language models for explaining the visual recognition system. We hope this study potentially advances interpretable AI and deepens our understanding of neural networks.

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
