# OpenReview forum: "GPT as Visual Explainer"
_ICLR.cc/2024/Conference — ICLR 2024 Conference Withdrawn Submission_

### Official Review · Reviewer_1ve1 · 2023-10-30

**Soundness:** 3 good
**Presentation:** 2 fair
**Contribution:** 3 good
**Rating:** 5
**Confidence:** 4

**Summary:**

This paper introduces an LLM-based explainability approach for vision models. It uses LLMs to explain different visual concepts contained in the image in a tree-like manner.
1) Given a predicted output from a vision model, LLM is used to explain predicted concept and its constituent parts using natural language
2) Text-to-image model is then used to identify the visual representations of the constituent parts.
3) These visual representations are then passed to the vision model for prediction.

Then steps 1 - 3 repeat.

This type of recursive procedure helps to explain visual concepts in a hierarchical tree-like structure.
The authors also propose to prune infrequent nodes and expand the tree based on LLM prompting. In addition to that the paper also proposes to retrain the model based on the refined explanation trees to improve the model’s interpretability. Plausibility, faithfulness, and stability are the metrics used to evaluate the explanations against baseline approaches.

**Strengths:**

The paper has a number of interesting contributions:

1) It uses a combination of a variety of models such as vision, text-to-vision and LLMs to generate tree-like explanations for the vision models.
2) Semi-automatically curates annotated datasets for CIFAR10, CIFAR100 and Imagnet.

3) Shows that the explainability-guided regularizer can help with both model explainability and accuracy.

**Weaknesses:**

1) The paper seems a bit crowded with different contributions that do not read coherently.
2) Overall it is known that all models( text2image, vision and LLMs) have prediction errors. In this case it will lead to error propagation in the tree of the explanations which can result in error amplification. It would be good to study the impact of the erroneous predictions on the explanation tree. E.g. how much the errors from LLM model get propagated down to text to image and vision models.
3) The abstract seems a bit too crowded and could be refined and simplified. For example the following sentence: `This tree is dynamically pruned and grown by querying the LLM using language templates, … `.
It is unclear what `language templates` is meant here.
4) Figure 1 is hard to interpret, the order of the arrows is not very clear. I’d recommend using numeric numbers on the arrows. This will help to better understand the sequence of the actions.
5) It’s unclear why the authors choose `Concepts, Substances, Attributes, and Environments.` attributes.
6) The explanation tree can potentially become very large and there can be different ambiguous cases. It would be good to discuss the problems and solutions related to scale and ambiguity. A discussion section can be helpful.
7) The same concept can be described in different ways through text. It would be interesting to study and discuss those aspects in the paper.
8) The evaluation part is a bit unclear. It would be good to clearly showcase the use cases (examples) where other baseline approaches fail and proposed method is able to handle those challenges better.

**Questions:**

1) How was the quality of the annotated dataset established ? Since it is semi-automated it can still have a high error rate or there might be many ambiguous cases. How are the ambiguities handled ?
2) In terms of calibration that leads to accuracy improvement is it total accuracy or class based accuracy ? Overall accuracy might be high but subgroups might perform poor.

---

### Official Review · Reviewer_dC7U · 2023-10-31

**Soundness:** 3 good
**Presentation:** 3 good
**Contribution:** 3 good
**Rating:** 6
**Confidence:** 3

**Summary:**

This paper aims to provide structured and human-understandable explanations for vision models and introduces a new and challenging task of generating visual explanatory tree.
This work collects data used for the explainability for the existing dataset ImageNet and CIFAR complementing the lack of hierarchy annotation.
The approach leverages LLM and text-to-image API as a bridge between language and vision domains.
This paper also introduces new benchmarks and metrics for assessing the quality of predicted tree-structured explanations.

**Strengths:**

Compared to previous explainability approaches, by leveraging the strengths of LLM, this method can construct abundant parsing tree used for the explanation of the visual models.
The building approach of the new dataset can automatically collect hierarchical annotations is significant.

**Weaknesses:**

As claimed as a work to generate human-understanable explanable parsing tree, this paper should include human evaluators results of assessing whether the generated results are reasonable. Without human judgment, these outputs cannot be properly evaluated.

**Questions:**

How does this model perform on out-of-domain categories? Can it still produce interpretable results if the category is not within ImageNet?